**Data Availability Statement:** The data that support the findings of this study belongs to University of

# Self-care practices and health-seeking behaviours in patients with dengue fever: A qualitative study from patients' and physicians' perspectives

Wei Leik Ng[1]*, Jia Yong Toh[1], Chirk Jenn Ng[1,2,3]*, Chin Hai Teo[1,4], Yew Kong Lee[1,4‡], Kim Kee Loo[1‡], Haireen Abdul Hadi[1‡], Abdul Muhaimin Noor Azhar[5‡]

**1** Department of Primary Care Medicine, Faculty of Medicine, Universiti Malaya, Kuala Lumpur, Malaysia, **2** Department of Research, SingHealth Polyclinics, Singapore, Singapore, **3** Health Services and Systems Research, Duke-NUS Medical School, Singapore, Singapore, **4** UM eHealth Unit, Faculty of Medicine, Universiti Malaya, Kuala Lumpur, Malaysia, **5** Department of Emergency Medicine, Faculty of Medicine, Universiti Malaya, Kuala Lumpur, Malaysia

☯ These authors contributed equally to this work.
‡ These authors also contributed equally to this work.
* wlng@ummc.edu.my (WLN); ngcj@um.edu.my (CJN)

## Abstract

### Introduction

Outpatient management for dengue fever is the mainstay of treatment for most dengue cases. However, severe dengue can develop rapidly while patients are at home. Understanding the self-care practices and healthcare-seeking behaviours among dengue patients managed as outpatients will help improve the delivery of care to these patients.

### Objective

This study aimed to explore the self-care practices, health-seeking behaviour and outpatient management of dengue fever from the perspectives of patients and primary care physicians.

### Methodology

This qualitative study used in-depth interviews and focus group discussions to obtain information from laboratory-confirmed dengue patients who received outpatient care and primary care physicians who cared for them. Patients and physicians shared their experiences and perceptions of self-care practices, decisions to seek urgent care, and outpatient management procedures and visit frequency. Data were coded and analysed using thematic analysis.

### Results

13 patients and 11 physicians participated. We discovered that the use of traditional remedies was common with patients perceiving no harm from it, whereas physicians did not see

Malaya Medical Centre (UMMC). Restrictions apply to the availability of these data, which were used under license for the current study, and so are not publicly available. Data can be requested from the UMMC ethics committee (email: ummc-mrec@ummc.edu.my).

**Funding:** The authors (WLN, JYT, CJN, CHT, YKL, KKL, HAH, AMNA) disclosed receipt of the following financial support for the research: This work was supported by the University of Malaya Impact-Oriented Interdisciplinary Research Grant Programme (grant number: IIRG002B-2020HWB, URL: https://umcms.um.edu.my/sites/research-cluster-office-v2/impact-oriented-interdisciplinary-research-grant-iirg-programme). The funders had no role in study design, data collection and analysis, decision to publish, or preparation of the manuscript.

**Competing interests:** The authors have declared that no competing interests exist.

a benefit. Dengue patients' knowledge of warning signs was inadequate despite the information being provided by physicians during clinical follow-up visits. Regarding the decision to seek urgent medical care, physicians assumed patients would seek help immediately once they experienced warning signs. However, for the patients, other factors influenced their health-seeking behaviour, such as their personal perceptions of symptom severity and often more importantly, their social circumstances (e.g., availability of childcare). Patients also described regular outpatient follow-up for dengue as inconvenient. There was variation in the prescribed outpatient follow-up interval recommended by participating physicians who complained about the lack of clear guidelines.

## Conclusion

Perceptions around self-care practices, health-seeking behaviour and outpatient management of dengue often differed between physicians and patients, especially on comprehension of dengue warning signs. Addressing these gaps between patient and physician perceptions and recognition of patient drivers of health-seeking behaviour are needed to improve the safety and delivery of outpatient care for dengue patients.

## Author summary

Dengue is a mosquito-borne viral disease endemic in many tropical and subtropical regions. Most dengue cases are self-limiting and can be managed as outpatients without hospitalisation. Once diagnosed, patients are advised to return to the clinic daily for monitoring until they recover. However, some patients can deteriorate rapidly at home in between follow-up and fail to recognise the deterioration. Poor self-management often leads to a delay in seeking help and treatment; this significantly impacts morbidity and mortality. This research provides insight into the self-care practices, health-seeking behaviour and outpatient management of dengue from the perspectives of patients with dengue and primary care physicians who managed them.

## Introduction

Globally, the incidence of dengue has increased in the recent few decades, with cases reported in up to 130 countries [1]. Americas, Southeast Asia, and Western Pacific regions are mostly affected, with Southeast Asia and Western Pacific region bearing 70% of the global dengue burden [2, 3]. It was estimated that worldwide, more than 9,000 dengue deaths occurred annually between 1990 and 2013 [1]. Southeast Asia, including Malaysia, reported some of the highest dengue mortality rates in the world [1]. Dengue fever has been endemic in Malaysia since the 1990s and incidence has increased dramatically from 32 cases per 100,000 in 2000 to 361 cases per 100,000 population in 2014 [4].

Outpatient management of dengue fever by primary care physicians is the mainstay of treatment as most dengue cases are mild and self-limiting [5]. Dengue is usually suspected when a high fever is accompanied by symptoms such as headache, retroorbital pain, muscle and joint pain, nausea, vomiting or rash [4]. There are three phases of illness in dengue fever: febrile, critical, and recovery. The critical phase typically comes after three days of fever and is marked by a rapid drop in body temperature; this may be misconstrued as a recovery although

patients may actually develop bleeding, rapid loss of fluid and organ damage during this phase [4, 5]. Once diagnosed with dengue fever, patients are usually expected to come back to the clinic daily to monitor clinical symptoms and complete blood count parameters to catch any deteriorating signs for early intervention [4]. Physicians usually watch out for warning signs for severe dengue during the follow-up, such as abdominal pain, persistent vomiting, bleeding gums or nose, bloody vomit or stool, breathlessness, lethargy and confusion [4]. They also look for changes in the complete blood count parameters such as increased haematocrit with concurrent rapid drop in platelet count [4]. Optimal outpatient clinical management is crucial to reduce morbidity and mortality in dengue infection.

Effective outpatient management depends on appropriate self-care practices and timely health-seeking, especially when the patients' condition deteriorates. Dengue patients managed as outpatients are usually advised to seek medical help immediately when dengue warning signs emerge. This usually occurs during the critical phase of dengue and patients with comorbidities such as diabetes mellitus and hypertension are at higher risk of developing severe dengue [6, 7]. Sometimes patients delay seeking help or miss their regular follow-up visits during the acute illness. Reasons for this include poor comprehension of health advice provided by physicians, poor self-care, and personal health beliefs [8–11].

There is a dearth of research on self-care practices and health-seeking behaviour of patients with dengue, especially in Southeast Asia; both depend on local socio-cultural contexts and health systems. In this study, we aimed to explore the self-care practices, health-seeking behaviour, and outpatient management of dengue fever from the perspectives of patients and primary care physicians. The findings from this study would give us an insight into how to improve health outcomes among patients with dengue in the outpatient setting, including promotion of rapid presentation to health facilities when patients experience warning signs.

## Methods

### Ethics statement

This study was approved by the University of Malaya Medical Centre Ethics Committee (MREC-20201021-9157). All participants provided written informed consent before enrolment in the study.

### Study design

This study employed qualitative research tools to explore patients' and physicians' experiences with dengue clinical management. A descriptive interpretive approach was used to describe and interpret the collective experience of the participants in the health setting [12].

### Participant recruitment

The participants of this study consisted of patients and primary care physicians at University of Malaya Medical Centre (UMMC). UMMC is in a central urban area in Malaysia with a high annual incidence of dengue.

The patients were identified from the Department of Primary Care Medicine, UMMC electronic medical records. Once identified, patients were screened for eligibility using the following criteria: i) ≥18 years of age, ii) serologically confirmed dengue fever case, and iii) received outpatient follow-up for their illness (dengue) within the past three months. The patients were selected purposively based on their age, ethnicity, and severity of dengue (not admitted; admitted) to achieve maximal variation in terms of their self-care needs and challenges. The physicians were identified from the primary care clinic at UMMC, with eligibility being having

experience in managing outpatient dengue cases. They were the family medicine trainees based in the clinic.

The eligible participants were contacted via phone and recruited by researchers or a research assistant. Appointments were set for the participants to be interviewed online via Zoom. The participants were reimbursed RM50 (approximately USD 12) for their participation in effort and time spent on the interview.

We stopped recruitment when we reached data saturation during data collection and analysis for patients' and physicians' interviews. Data saturation was reached when no additional responses were found from patients' and physicians' interviews, and no new themes emerged. One to two additional interviews were then conducted for patients and physicians to ascertain data saturation.

## Data collection instruments and activities

Separate interview guides for patients and physicians were developed based on health behaviour theories, namely the health belief model [13] and theory of planned behaviour [14], and literature review. The authors reviewed 13 articles focusing on the health belief, attitudes, practices, and experiences related to dengue fever to help develop the interview guides (S1 and S2 Interviews). They were used to facilitate the semi-structured interviews. During the interviews, physicians were asked to describe their experiences when treating dengue patients, challenges that they encountered during the consultation and subsequent follow-up with dengue patients, their practice in dengue case management and experiences in handling difficult dengue cases. In contrast, patients were asked to recall their experiences throughout the illness, from initial presentation, diagnosis, follow-up till recovery, and describe their experiences during consultations with physicians, self-management practices at home, and health service utilization. In addition, patients were asked to provide suggestions to improve their healthcare experiences. In-depth interviews (IDIs) were conducted for patients while focus group discussions (FGDs) and in-depth interviews were used for physicians.

Interviews were conducted by WLN, JYT and CJN. WLN and CJN were clinicians and medical lecturers in Department of Primary Care Medicine, UMMC. JYT was the research assistant for the study. WLN, CJN and JYT held medical degrees. Additionally, CJN had a PhD and master's degree in family medicine while WLN had a master's degree in family medicine. Interviews were conducted in either Malay or English, depending on the participants' preferred language as all researchers were fluent in both languages.

Due to the COVID-19 pandemic, the study was conducted using the online platform Zoom (Zoom Video Communications, California). To ensure confidentiality, all teleconference sessions were only accessible using a password. The participant information sheet and electronic consent form (Google Form) for participation were sent to the participants through email/ WhatsApp, and the consents were obtained before the interview. Participants were allowed to turn their video off if they were uncomfortable keeping it on. All interviews were recorded using Zoom's function after obtaining consents. No one else besides the participant and researchers (WLN, JYT and CJN) was present during the interview. Researchers took notes during the interviews. The mean duration of the IDIs (range: 21 to 55 minutes) and FGDs (range: 51 to 55 minutes) was 36 minutes and 53 minutes, respectively.

## Data analysis

The audio recordings were transcribed verbatim and anonymised. The transcripts were analysed independently by two researchers (WLN and JYT) using the thematic approach, which was first by reading and re-reading the transcripts to familiarise themselves with the content,

followed by open coding and subsequently axial coding. Triangulation was done by comparing the findings between the physicians and patients to identify similarities or differences in their perceptions. The researchers met to compare and discuss the coding and reconcile any discrepancies via a team discussion. The researchers critically examined and reflected on their roles throughout the study to reduce potential biases during the interviews and data analysis. ATLAS.ti version 8 software was used to manage the data. Transcripts were analysed in either Malay or English as all researchers were familiar with these languages; participant quotes in Malay were translated into English for reporting.

## Results

13 patients were interviewed by IDIs and 11 physicians by FGDs and IDI (two FGDs consisting of five physicians per session; one doctor via IDI). Five patients did not pick up our phone calls during the recruitment phase, while another three patients refused to participate after we contacted them. We did not pursue the reasons for refusal. None of the physicians we approached refused participation. These interviews were conducted between 8 December 2020 and 5 June 2021.

Most patients were working adults with no comorbidities. All of them had received outpatient follow-up for dengue fever, and a few developed clinical conditions that required hospitalisation during their illness. All interviewed physicians were primary care physicians with experience managing dengue fever. The demographic details of the study sample are shown in Table 1.

Four themes emerged from our interviews with patients and physicians.

### Theme 1: Use of traditional remedies for dengue

Dietary modification was the main change during self-care at home. Increasing fluid intake was the main modification made. The modification in fluid intake was primarily influenced by

**Table 1. Demographic summary of the participants.**

| Patients (N = 13) | |
|---|---|
| **Parameters** | **Description** |
| Age | 19–58 years old (mean age: 35, standard deviation: 13) |
| Gender | 9 males, 4 females |
| Ethnicity | 5 Bumiputera, 5 Chinese, 2 Indian, 1 Syrian |
| Comorbidities | 10 without comorbidities, 3 with comorbidities such as hypertension, diabetes mellitus, gout and iron deficiency anaemia) |
| Hospital admission | 4 were admitted for dengue while 9 were not. |
| Mean time elapsing between the interview and onset of dengue. | 46 days (range: 8 to 77 days) |
| History of dengue (prior to current episode) | Only two participants had previous episodes of dengue, three and five years prior to their current illness. They only received outpatient care with no hospitalisation during the previous episodes. |
| **Physicians (N = 11)** | |
| **Parameters** | **Description** |
| Age | 32–35 years old (mean 34, standard deviation 1) |
| Gender | 6 males, 5 females |
| Ethnicity | 1 Bumiputera, 8 Chinese, 2 Indian |
| Years of practice | 5–10 years |
| History of dengue fever | None had ever been personally infected with dengue fever |

physicians' advice. The fluid choice varied from warm water, lemon water, isotonic drink, coconut water, soup, or barley water.

*"The doctor informed me, ok for dengue all I need is to drink more water, there is no cure for dengue, ok, since there is no cure and then good enough. . ." (Patient 6, 58-year-old male, not admitted)*

Traditional remedies such as papaya leaf, guava juice and crabmeat soup were commonly used by patients; papaya leaf was the most frequently mentioned. Our patients consumed papaya leaf in the form of juice (boiling water together with papaya leaf) or took pills made from extract. Papaya leaf juice was more commonly ingested. The use of traditional remedies was influenced by family, friends, community (neighbour) and social media. Patients perceived no harm from taking them, regarding them as part of their usual diet. Patients generally heeded the advice from their physicians to increase their fluid intake but took traditional remedies to complement the fluid. Patient 3 also mentioned that the desperation to get well again motivated her to try traditional remedies, knowing that there was no definitive treatment for dengue. She also shared an anecdote she heard from others that papaya leaf could help boost immunity in dengue.

*"They also made like, you know, moms being moms, they make a weird type of soup for you to drink, like crab soup I don't know what the hell is that? So basically, that and I also, I guess they also heard from other people that, you know, if you are having dengue, you have very weak immune system, you should be taking things like papaya extract pill." (Patient 3, 25-year-old female, not admitted)*

*"I only believe in like scientific evidence, obviously, I don't believe in all of these. But I guess like, you know, when you are desperate, you just want to try everything to be well again, I mean, there is no cure for dengue." (Patient 3, 25-year-old female, not admitted)*

Some patients received advice from physicians not to take such traditional remedies. Patient 1 commented that his doctor did not address any dietary measures apart from increasing fluid intake.

*"The doctor asked me to drink (plain water), not to drink papaya leaf juice. She told me not to do that, because it was very common among the patients. But my parents ended up giving it to me, but I did, I said I didn't want to, but I still drink it anyway." (Patient 5, 20-year-old male, admitted)*

*" They (physicians) didn't mention eating, they didn't tell me eat this, don't eat that. Nothing, just drink water, that's it." (Patient 1, 30-year-old male, not admitted)*

*"Recently I got admitted, the specialist, he mentioned to me don't drink crab soup, papaya juice. And then I also kept quiet, don't know the reason, because they are normal food only, to me they are not any medicine" (Patient 13, 35-year-old female, admitted, had previous history of mild dengue)*

When asked if physicians gave them any reason not to take traditional remedies such as papaya leaf and crabmeat soup, lack of scientific evidence was the reason quoted by physicians.

*"Two weeks later, when I follow up with doctor (a different doctor), I asked the doctor, "Why did the specialist stop me from drinking crabmeat soup?"*

*After that, he answered, "Maybe because there's no research that showed crabmeat soup helps with dengue."*

*Okay, I am okay with that, that's what I understand." (Patient 13, 35-year-old female, admitted, had previous history of mild dengue)*

The physicians in our interview generally took a more middle-ground approach when it came to traditional remedies. They perceived no benefits from the additional dietary measures but did not explicitly forbid patients from taking them. Doctor 3 would not stop her patients from taking these remedies in the fluid form, provided that the remedies were not some medicines. Physician-patient relationship was also a consideration. Doctor 1 mentioned that it is important to respect the patients' belief in traditional remedies provided it caused no harm, to maintain a good physician-patient relationship. He further explained that if physicians did not respect them, it would cause a barrier between them, making it difficult to develop trust with patients for future management.

*"I'll ask what they take. If as long as I think it is safe, and it has some fluid inside it, like soup, papaya juice, because all these I will tell them that definitely from research, it is not proven. But as long it is not harmful for them, and they are drinking something, so I would say, I would be clear to them, I would say it doesn't have any beneficial thing, but at the same time if let say if it's like harmful or anything then I will stop them from taking." (Doctor 10, male, 6 years of experience)*

*"If they can't drink water, drink whatever fluid that is available, as long as it is juice, juice is okay. As long as it is fluid, but definitely not traditional medicine." (Doctor 3, female, 10 years of experience)*

*"I mean the behaviour is not harmful for them, we should respect, because, if we want to change one person behaviour, then it might affect the relationship between us with the patient, and is hard for us to continue follow-up with them. . . . But then once we change their perception on this traditional medication we said it might cause harm to you, then their relationship with us will start to have some barrier then." (Doctor 6, male, 5 years of experience)*

Doctor 11 pointed out that she did not encourage the intake of traditional remedies. She did not want to give a wrong impression to her patients that the traditional remedies were the substitute for self-monitoring of warning signs and adequate fluid intake.

*"So I don't actually encourage, but if they want to take it, I don't do anything. Basically, I don't encourage. I don't want to give the point for them to just take it (traditional remedy) without taking care of other factors that are more important like the warning signs, the fluid intake everything." (Doctor 11, female, 9 years of experience)*

### Theme 2: Gaps in information provision and patients' knowledge

There were gaps in the patients' knowledge of the dengue warning signs and different phases of illness. When specifically asked about the warning signs of dengue they watched out for during their dengue fever episode, most patients mentioned vomiting, bleeding, and diarrhoea. Some participants did not remember receiving any advice on warning signs of dengue. Only one participant mentioned dengue fever causing death when asked about her understanding of the severity of dengue fever.

*"Ya, they told me the warning signs, ya, like diarrhoea, bleeding, if throwing up, ya, I have to go to hospital. But other than that, no, just drink water." (Patient 1, 30-year-old male, not admitted)*

*"When entering critical phase I was shocked, because I did not have fever, and then doctor told me I was actually in the critical phase. We thought if we stay at home, we feel no more fever, we are getting better." (Patient 13, 35-year-old female, admitted, had previous history of mild dengue)*

Patient 7 did not seek immediate treatment despite persistent vomiting because she felt she had nothing serious, based on what the doctor told her during her first visit to clinic.

*"I kept on vomiting every five minutes. . .. I just rest, I just keep drink a lot of water. But after I drink water, I still vomit, the water out, yeah. But I just have to wait for the next day. . ... .. because I went to the clinic (on earlier visit) and then the doctor said I have nothing serious. So, I didn't go to the hospital." (Patient 7, 19-year-old female, admitted)*

Our physicians tailored the information on warning signs and dengue phases provided to their patients, based on their perception of the patients' existing knowledge of dengue and their ability to understand. Doctor 4 would explain the different dengue phases to her patients, while Doctor 3 thought explaining dengue phases to patients was more than they could comprehend or retain. Doctor 11 felt that most patients already knew about dengue before seeing physicians.

*"Usually once confirmed with dengue, I will show them the dengue phase chart, then it would show like the febrile phase, the critical phase and also the recovery phase. I will let them know exactly what phase they are in, and then the timing of what to watch out for, the critical phase and all that, and then would inform them about the importance of monitoring with the CBC (complete blood count)." (Doctor 4, female, 10 years of experience)*

*"I didn't go to the whole three phase thing; I think it's a bit too much for the patient. So, I'll just state what you need to look out for, all the warning, there's only a few warning signs so it's easy to remember." (Doctor 3, female, 10 years of experience)*

*"I think most of our patients have at least a bit information about dengue even before they came to see the doctor, they already know that there are some sorts of dengue haemorrhagic fever, something like that, they know few things already. We just emphasise the danger of it" (Doctor 11, female, 9 years of experience)*

## Theme 3: Symptom severity and social circumstances influence health-seeking

Our patients' decision to seek immediate medical care was influenced more by the severity of symptoms they experienced and their social circumstances rather than their knowledge of the warning signs. Patient 2 postponed his plan to visit the hospital even though he was feeling unwell (severe dizziness) because he had no one to take care of his children while he sought care. Patient 11 also cited not having available childcare as the reason for not seeking emergency care.

*"I really wanted to go, hospital. But nobody was at home to take care of my children that time. My wife was working." (Patient 2, 36-year-old male, not admitted)*

*"When I was diagnosed with dengue, my family from my hometown. . .they came to my house, they took my two children back to my hometown. So, I was not looking after my children at that time, and I am with my wife only. My wife can take care of me and send me to hospital daily. But for others, it may be difficult for them to come to hospital if there is no one to send them, especially on weekends where there are no babysitters." (Patient 11, 29-year-old male, not admitted)*

Perceived severity was based more on how badly individual patients felt (e.g. pain level, general health status) rather than on recognition that they had one of the warning signs (provided by physicians during patient visits). For example, patients would go to the emergency department immediately if their headache or retroorbital pain exceeded their own tolerance levels. Another patient went to the emergency department due to unbearable gastric pain, but he did not recognise this as a dengue warning sign.

*"Maybe if my headaches or my eyeballs had been really painful, I would know something is not right. But if the pain is only very mild, then it was not a necessity for me to go to hospital. I mean it's our own body, we know how serious it is" (Patient 6, 58-year-old male, not admitted)*

How the patients decided to seek immediate care differed from our physicians' assumption that patients already had the necessary knowledge to decide when to seek immediate help. Physicians generally advised patients to seek treatment when they developed warning signs. Doctor 2 said he did not face any resistance from patients when he advised them to go to the hospital immediately if they had warning signs, because they were familiar with them.

*"Usually, no resistance from most of the patients, because seems like everyone has like the same baseline knowledge about dengue, and it's quite acceptable that level of knowledge." (Doctor 2, male, 7 years of experience)*

## Theme 4: Dengue follow-ups visits were 'troublesome' to patients

When asked about outpatient management of dengue fever, patients did not appear to understand the reasons for regular, daily follow-up in primary care clinics. Some patients questioned the rationale of regular follow-up when they felt they were able to assess their condition themselves. Concerns about daily follow-up visits included their financial cost (medical expenses), dislike of hospital visits in general, feeling physically fine, dislike of venepuncture, logistic hassles, and long waits (hours) in the clinic.

*"I'm not too sure what is the frequency of the visit. So, I just follow advice by the doctor" (Patient 10, 47-year-old male, not admitted)*

*"Yes, I was thinking why I can't come back like maybe every two days or more like that. I only come back daily if like you know, they tell me the red flags that I should be aware of like ok maybe if you get this, then you come back, you know, but then otherwise maybe after two days you repeat blood again. You know, I thought it could have been done like that." (Patient 3, 25-year-old female, not admitted)*

*"And also, the cost because I am paying also, I am paying a lot. So, like I feel I am ok, I feel my body getting better, why do I have to go and pay every day that amount of money." (Patient 1, 30-year-old male, not admitted)*

"*To me the very troublesome and cumbersome thing was having to go back every day for the blood test. I was actually thinking why I can't come back like maybe every two days or more*" (Patient 3, 25-year-old female, not admitted)

"*Of course, taking a blood test is not pleasant. You must stick a needle through your skin, it's not pleasant.*" (Patient 5, 20-year-old male, admitted after three days of outpatient follow-up)

Lack of continuity of care in a public government clinic was also one of the factors mentioned by patients as the reasons they found daily follow-up for dengue unpleasant.

"*That's why I decided to go to clinic (private general practitioner clinic). Because at least you see one doctor. I don't wanna see a different doctor every day, because I need to tell them what happened every time, they ask the same questions. You should read the report, know what happened to me. I am not gonna tell you every time.*" (Patient 1, 30-year-old male, not admitted)

The follow-up visit interval recommended by the physicians to their patients varied considerably. Although daily follow-up visits are recommended by most international guidelines, our physicians often adjusted the follow-up visit intervals, depending on patient comorbidities, the patient's current dengue phase (febrile, critical or recovery), if the patient had a serologically confirmed dengue diagnosis, the patients' clinical and vital signs, history of past dengue infection, and distance from the hospital. The physicians expressed that the national clinical practice guidelines did not adequately address the issue of outpatient follow-up visit frequency. They felt there was no clear-cut guide on monitoring dengue patients as outpatients apart from watching out for warning signs. Outpatient management of dengue patients by physicians were guided by personal clinical experience, phase of dengue illness and clinical parameters. Normally, the physicians followed up their dengue patients daily but would space out the follow-up at variable intervals when patients were in recovery phase.

" *I don't think it is stated in the dengue CPG (clinical practice guideline) like how frequent we need to monitor. What I could recall is from my time as an intern, but this varies by hospital or clinic where you were trained, and we still follow that until today.*" (Doctor 1, female, 10 years of experience)

"*It depends on which stage the patient is in. So, let's say if the patient is in febrile phase, then we might be seeing them more often. But if the patient is already in recovery phase, and then the platelet is ok, more than 100 then maybe will see them after more time, maybe 5 days after, 1 week.*" (Doctor 7, male, 7 years of experience)

Generally, the physicians in our study thought most patients were compliant with regular follow-up because they understood the severity of dengue fever and were afraid of complications.

"*I think when comes to dengue I think they are a bit worried, so I think they are compliant when it comes to dengue, when I'll tell them about the complications of dengue, what can happen. So, when you tell them about complications and you include death, they will be a bit careful on that*" (Doctor 10, male, 6 years of experience)

Weekends influenced physicians' recommendations for follow-up visits. Physicians would be more prone to skip a day or two of monitoring over the weekend if patients were stable,

especially in recovery phase. Discontinuity of care and rejection by emergency departments (EDs) were cited as problems associated with weekend monitoring of outpatient dengue cases.

> "*Yeah, especially towards the weekends. Like when they come over on Friday and they need to continue daily monitoring on Saturdays and Sundays. And there are changes to where they can go to. That becomes a bit difficult for us cause sometimes you say come to ED, but then ED does not accept them, so that becomes a hassle for us to find out and also for the patients because they don't know where to go.*" (Doctor 2, male, 7 years of experience)

> "*Sometimes when I think the CBC (complete blood count) was quite stable, and the patient was quite stable, sometimes because of the weekend delay, I just proceed to Monday*" (Doctor 1, female, 10 years of experience)

Some patients also brought up the weekend factor concerning childcare. As mentioned in the quote by patient 11 in Theme 3: Symptom severity and social circumstances influence health-seeking, parents might find it harder to look for babysitters during the weekend, and consequentially, making it difficult for them to come for follow-up during the weekend.

## Discussion

Four themes regarding the self-care practices and outpatient dengue management emerged from patient and physician interviews. They were 1) use of traditional remedies for dengue, 2) gaps between the information provided by physicians and patient knowledge, 3) the influence of symptom severity and social circumstances on health-seeking, and 4) the perception among dengue patients that follow-up visits were 'troublesome'. Although the perspectives of both patients and physicians had some overlap, it was clear that the two groups had distinct perspectives on self-care practices, motivations for seeking hospital care and outpatient management of dengue fever.

Perceptions related to the use of traditional remedies were quite distinct between patients and their treating physicians. When patients described using unproven traditional remedies to treat their dengue symptoms, they were often discouraged by their physicians who expressed concern that remedies without scientific evidence of benefit would send the wrong message to patients. Physicians expressed concern that focusing on traditional remedies might distract patients from taking adequate fluid and the need to monitor themselves for warning signs (see comments from Doctor 11). Interestingly, many traditional remedies came in the form of liquids and theoretically would assist patients ingest sufficient liquids. This issue created some tension among physicians and patients, especially when patients often used traditional remedies against physician recommendations because they saw no harm in doing so.

A plausible clinical approach to the use of traditional remedies, as expressed by some of our physician participants, would be to take the 'middle ground' stand, where physicians can acknowledge that these remedies cause 'no harm' but do not provide additional benefit beyond the need to ingest liquids. Although these remedies do not have specific physiological benefits, for example they do not enhance the immune response, patients can be counselled to decide for themselves on the traditional remedies, with emphasis on ensuring adequate fluid intake and monitoring themselves for warning signs. This may represent a subtle change in messaging from physicians, where there is recognition that traditional remedies may have some coincidental benefit of increased fluid intake. Secondly, it is important to explain to patients that the symptoms they experience are because their immune system is functioning, not because of a "weak" immune system. This will discourage the intention to use remedies in the form of pills to boost immune system.

Physicians should discuss traditional remedies and practices with their patients, especially for the case of papaya leaf extract, which has been promoted in many dengue endemic countries, especially India, Indonesia and Malaysia [15]. While there is limited evidence suggesting that papaya leaf extract improves platelet counts, this may not translate to clinical improvement [15, 16]. A larger concern associated with this product is that it may encourage self-treatment and discourage people from seeking health care. More than 80% of patients with dengue fever in Malaysia used some form of traditional medicine [17, 18]. Liu et al. (2020) showed that up to 40% of the patients with fever who suspected dengue fever practiced self-medication and traditional medication without seeking treatment in a hospital [19]. In this scenario, infected people may not recognize warning signs, and in the case of severe dengue, present to emergency care when it is too late, resulting in mortality.

Additionally, physicians need to understand the factors that influence patients to consider alternative treatment. The treatment for dengue is not curative but mainly supportive (alleviation of symptoms and fluid therapy). Patients may perceive that the modern treatment for dengue is ineffective and not definitive, driving them to look for alternative treatments [11, 20]. Physicians must emphasize that their patients ingest lots of liquids, monitor their condition and come to the hospital quickly if they exhibit warning signs. The message is that their condition can deteriorate just as easily with traditional remedies as without. Finding ways for patients to recognize warning signs and respond to them rapidly by seeking hospital care is an important gap identified by this study.

Although the physicians in this study all discussed warning signs and their importance with their patients and felt confident that their patients were adequately informed, many of the participating patients either did not remember or adequately understand these messages from their physicians. This occurred despite patients coming regularly for their follow-up visits where information on warning signs and stages of dengue fever were reinforced. A previous study from Malaysia also demonstrated that the general public could identify the common symptoms of dengue fever such as high fever, headache, arthralgia and myalgia but has more difficulty recognizing warning signs such as rapid breathing, restlessness and severe abdominal pain [21]. We found some perceptions between physicians and their patients did not match. Physician responses indicated that they felt that the patients' knowledge about dengue and warning signs were adequate contrary to the information provided through patient interviews. All the participating physicians emphasized information on warning signs to their patients and provided similar information in a written note for patients to take home. However, it was clear that this information was not retained with some patients unable to recall any warning signs provided by their physicians. Several factors could have led to failure to transfer knowledge from physicians to patients, such as patient distress, poor health literacy and misinformation [22]. The gaps between information provision by physicians and knowledge retention by patients in our study highlight an urgent need for more effective communication strategies for outpatient dengue care.

When it came to the decision to seek immediate or urgent care, we noticed that what influenced patients to seek help (symptom severity and social circumstance) differed from the physicians' assumption (presence of danger signs). There was a lack of studies investigating the factors influencing decision to seek immediate care among outpatients with confirmed dengue fever. However, a recent systematic review of factors influencing healthcare-seeking in patients with dengue revealed findings like what we uncovered in our interviews. Among the similar factors we discovered included individual (such as personal health belief, risk perception), interpersonal (social support) and organisational factors (access to healthcare: inconvenient operating hours, long waiting time, oversaturated emergency services) [23]. Considering these other factors is essential in physicians' consultation with patients as the information conveyed

by physicians would be more beneficial if they addressed the patients' concern, belief and individual circumstances [24]. Physicians sometimes do not identify what patients consider important, potentially leading to varied healthcare-seeking behaviours among patients managed as outpatients for dengue fever.

Of note, childcare responsibility was one of the barriers to healthcare seeking found in our study. Childcare responsibility was widely recognised as a barrier to healthcare access, particularly among women [25–27]. Nair SK (2018) reported that patients, especially women, with infectious diseases such as dengue did not access healthcare services despite the doctor's recommendation, owing to familial responsibility [27]. Interestingly, both patients who mentioned childcare issues in our study were men, in contrast to the literature. Conventionally, men were seen as the breadwinners while women were the homemakers. Men were perceived to have more time to care for themselves and better healthcare access. For our sample, this may indicate a changing gender role and sharing of childcare responsibility among parents, especially in an urban setting like ours. Nevertheless, the neglect of proper care for acute illnesses such as dengue, which can deteriorate rapidly, is worrying. Setting up a non-profit childcare centre at the healthcare centres to address the lack of childcare is an interesting endeavour, as demonstrated by Alvarez KS et al (2022) [28]. This would require upgrading facilities, integrating childcare service into the clinic workflow, and engaging community-based childcare organisations [28].

In this study, we also uncovered concerns among patients and physicians regarding the regular follow-up for dengue fever. Daily or more frequent follow-up is recommended for patients receiving outpatient care for dengue fever [4, 5]. Our patients did not appreciate the significance of daily follow-up to monitor for warning signs, particularly during the critical phase. Compliance with the regular follow-up was still there mainly due to adherence to the doctor's instruction. We also discovered organisational issues related to daily follow-up that can be improved in the primary care setting: long waiting time, discontinuity of care and inadequate operating hours, particularly over the weekend. The weekend factor needs to be tackled as it also influences how our physicians dictate the frequency of outpatient follow-up for dengue patients. Similar to previous reports, our physicians practised an instinctive clinical approach to manage outpatients based on individual judgement and experiences, citing the inadequacy of current guidelines [29]. WHO guidelines recommend "daily follow-up for all patients except those who have resumed normal activities or are normal when the temperature subsides" [30]. Local Malaysia dengue guidelines also recommend daily monitoring during the febrile phase (more frequently towards late febrile phase), at least twice daily during critical phase and daily (or more frequently as indicated) during the recovery phase [4]. The practices we observed in our interviews with physicians did not conform to these guidelines. The practicality of asking patients to come twice to the clinic for follow-up in the same day is also questionable. We recommend a more pragmatic approach to decide on the frequency of follow-up to be included in the guideline, tailored to the clinical phases of dengue, individual circumstances, and healthcare setting. More research is needed to explore newer ways to manage outpatient dengue better. The concept of remote monitoring for dengue such as virtual consultation or mobile application for self-monitoring is interesting, with more data needed on its safety and efficacy.

## Strength and limitations

Challenges associated with outpatient care for patients with dengue fever represent a significant public health problem in all countries where dengue is endemic. We also explored this topic from both the perspectives of physicians and patients, allowing us the opportunity to

compare the similarities and differing perceptions between these two groups. Our findings are applicable beyond Malaysia and illustrate the need for alternatives to monitoring patients daily through hospital or clinic visits. While the importance of promoting awareness of dengue warning signs is recognized worldwide, there is a need to assess whether the message is communicated effectively to different populations.

Our study was carried out in a single a single health centre in an urban area in Malaysia representing more affluent setting, but many of the issues raised should cross socioeconomic levels and different cultural contexts. There may also be recall bias among our participants as some of the interviews were not conducted immediately after their illness.

## Conclusion

Perceptions around self-care practices, health-seeking behavior and outpatient management of dengue often differed between physicians and patients, especially in the comprehension of information about dengue warning signs. We need to rethink our health communication strategies on the traditional remedies, dengue warning signs and timely health-seeking in dengue. Addressing these gaps between patient and physician perceptions and recognition of patient drivers of health-seeking behavior are needed to improve the safety and delivery of outpatient care for dengue patients.

## Supporting information

**S1 Interview. Interview guide for physicians.**
(DOCX)

**S2 Interview. Interview guide for patients.**
(DOCX)

## Author Contributions

**Conceptualization:** Wei Leik Ng, Jia Yong Toh, Chirk Jenn Ng, Chin Hai Teo, Yew Kong Lee, Haireen Abdul Hadi, Abdul Muhaimin Noor Azhar.

**Data curation:** Wei Leik Ng, Jia Yong Toh, Chirk Jenn Ng, Chin Hai Teo.

**Formal analysis:** Wei Leik Ng, Jia Yong Toh, Chirk Jenn Ng, Chin Hai Teo, Yew Kong Lee, Kim Kee Loo, Haireen Abdul Hadi.

**Funding acquisition:** Wei Leik Ng, Chirk Jenn Ng, Chin Hai Teo, Yew Kong Lee, Haireen Abdul Hadi, Abdul Muhaimin Noor Azhar.

**Investigation:** Wei Leik Ng, Jia Yong Toh, Chirk Jenn Ng, Chin Hai Teo, Kim Kee Loo.

**Methodology:** Wei Leik Ng, Jia Yong Toh, Chirk Jenn Ng, Chin Hai Teo, Yew Kong Lee, Haireen Abdul Hadi, Abdul Muhaimin Noor Azhar.

**Project administration:** Wei Leik Ng, Chirk Jenn Ng, Chin Hai Teo, Kim Kee Loo.

**Resources:** Wei Leik Ng, Chirk Jenn Ng, Chin Hai Teo.

**Supervision:** Wei Leik Ng, Chirk Jenn Ng, Chin Hai Teo.

**Validation:** Chirk Jenn Ng.

**Visualization:** Wei Leik Ng, Jia Yong Toh, Chirk Jenn Ng, Chin Hai Teo, Yew Kong Lee, Kim Kee Loo, Haireen Abdul Hadi, Abdul Muhaimin Noor Azhar.

**Writing – original draft:** Wei Leik Ng, Jia Yong Toh, Kim Kee Loo.

**Writing – review & editing:** Wei Leik Ng, Jia Yong Toh, Chirk Jenn Ng, Chin Hai Teo, Yew Kong Lee, Kim Kee Loo, Haireen Abdul Hadi, Abdul Muhaimin Noor Azhar.

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
