## [Decision Letter · Decision Letter 0]

29 Jan 2023

Dear Dr. Ng,

Thank you very much for submitting your manuscript "Self-care practices and health-seeking behaviours in patients with dengue fever: a qualitative study from patients’ and doctors’ perspectives" for consideration at PLOS Neglected Tropical Diseases. As with all papers reviewed by the journal, your manuscript was reviewed by members of the editorial board and by several independent reviewers. The reviewers appreciated the attention to an important topic. Based on the reviews, we are likely to accept this manuscript for publication, providing that you modify the manuscript according to the review recommendations. 

As this is a qualitative study I am asking that the authors focus on addressing the recommendations of Reviewer's 2 and 3. I only ask that you consider Reviewer 1's comments in the context of an assessment from a strong clinical background and how you might reach an audience who is less familiar with qualitative work. The Instruments could be included as supplemental information which would address one major concern from Reviewer #1 and is consistent with Reviewer 2 and 3's request for more details on the interview methodology. Overall the issues are quite addressable and I agree the article would be of interest to a PLOS NTD audience.

Sincerely,

Amy C. Morrison, PhD

Academic Editor

Elvina Viennet

Section Editor

As this is a qualitative study I am asking that the authors focus on addressing the recommendations of Reviewer's 2 and 3. I only ask that you consider Reviewer 1's comments in the context of an assessment from a strong clinical background and how you might reach an audience who is less familiar with qualitative work. The Instruments could be included as supplemental information which would address one major concern from Reviewer #1 and is consistent with Reviewer 2 and 3's request for more details on the interview methodology. Overall the issues are quite addressable and I agree the article would be of interest to a PLOS NTD audience.

Reviewer's Responses to Questions

**Key Review Criteria Required for Acceptance?**

**Methods**

-Are the objectives of the study clearly articulated with a clear testable hypothesis stated?

-Is the study design appropriate to address the stated objectives?

-Is the population clearly described and appropriate for the hypothesis being tested?

-Is the sample size sufficient to ensure adequate power to address the hypothesis being tested?

-Were correct statistical analysis used to support conclusions?

-Are there concerns about ethical or regulatory requirements being met?

Reviewer #1: The audience for this work would be extremely limited: as the health practices and societal beliefs presented in this paper would be mostly limited to Singapore medical practice, with limited generalizability beyond.

The methodology is also limited: limited number of participants and practitioners took part. There is no explanation of typical qualitative best practices such as evidence of interview saturation or conceptual overlap, rendering the findings thoroughly acquired and relevant.

Reviewer #2: Minor revisions suggested - Within the "data collection and instrument" section a literature review is mentioned very briefly, some additional information surrounding the literature review would be beneficial e.g., was this conducted by the authors? How many papers were reviewed? A bit more information about the interviews would also be good. It is stated the interviews were conducted on Zoom, were these with video on or off? Singular tense is used for the clinician interview, does this mean only one of the clinicians were interviewed and the remaining participated in the focus group? Would be good to clarify. Other than these points I think the study design (including the objective) was clear, the population well described and analyses deemed appropriate.

Reviewer #3: Methods and study population are appropriate. Data analysis process was clearly explained. No ethical concerns.

a. Participant recruitment: How was the number of patients interviewed determined? How many patients and doctors refused to participate and what were the reasons for non-participation?

b. Data collection and instrument, pg. 6: How long on average did the individual interviews and focus groups last? Did all 11 physicians participate in both IDIs and FGDs?

**Results**

-Does the analysis presented match the analysis plan?

-Are the results clearly and completely presented?

-Are the figures (Tables, Images) of sufficient quality for clarity?

Reviewer #1: Limited results presented; limited relevance to perhaps only dengue treatment and care within Singapore.

Reviewer #2: Minor revision suggested - I would like some more information about the sample. Specifically, it would be good to know how much time had it been since patients had experienced dengue. As noted in the methods section patients had to have received outpatient follow up for dengue within the past three months - but what was the actual amount of time since experiencing dengue of participants? Some extra descriptive statistics of demographic characteristics would also be useful e.g., Mean age. I thought the results themselves were presented really well and told a comprehensive narrative.

Reviewer #3: Findings are clearly presented.

a. Nice integration of the qualitative data in the Results section. 

b. Pg. 8, Table 1

- Were patients asked about their history of dengue fever infection? Health seeking behaviour and risk perception may be affected by a previous, mild case of dengue as patients may underestimate their risk for severe dengue and overestimate the effectiveness of home remedies in resolving dengue fever symptoms.

- Suggest changing “Dengue Admission” to “Hospital Admission” since the parameter is whether or not the individual was admitted to the hospital.

c. Pg. 9, Quotes 

- Could the sex of the patient be added as well, e.g., “Patient 3, 25-year-old female, not admitted”. It can be useful to see how men and women discuss the topics. 

- It would be nice to have the sex of the physician as well, but only if it does not increase the possibility that the physician participant could be identified as there is background information included in the manuscript (i.e., all are primary care physicians training in family medicine and years of practice are listed after the quote).

d. Theme 1: Use of complimentary diet for dengue 

- Pg. 9, line 181: “leave” should be “leaf” 

- Pg. 10: Were the patients asked if the doctor gave them a reason why they should not drink papaya leaf juice or crab soup? While some of the doctors stated that they might tell their patients there was no scientific evidence for a complementary diet, the quote from “Patient 13, 35 years old, admitted” (“And then I also kept quiet”) is a perfect example of how doctors shut down communication with patients. 

- Pg. 10, line 214: Suggest not mentioning “CAM” as complementary and alternative medicine is much broader than the health practices and health seeking behaviours that are the focus of this manuscript. If it is part of a direct quote, then put the quote in so you don’t have to describe CAM in the introduction.

e. Theme 3: Symptom severity and social circumstances influence health-seeking 

- Did patients mention cost as a barrier to follow-up visits? Costs could be financial (transportation, medical expenses not covered by the health system) and time away from other things such as work, school, family, etc.

- Pg. 13: Not having someone to take care of children while the caregiver goes to the clinic has been identified as a major barrier to health seeking in general, especially for women, yet there is little in the dengue literature that addresses this issue. The authors could more fully discuss this issue in the Discussion. While not ideal, would video visits be an option for some of the patient follow-up care?

- Pg. 16, line 353: Define “FBC” in quote.

**Conclusions**

-Are the conclusions supported by the data presented?

-Are the limitations of analysis clearly described?

-Do the authors discuss how these data can be helpful to advance our understanding of the topic under study?

-Is public health relevance addressed?

Reviewer #1: The questionnaire/discussion guide was not shared--making conclusions that the authors came up with hard to interpret.

Several "conclusions" seemed to be perceptions of the authors, rather than necessary fact from the qualitative work.

Reviewer #2: The conclusions are supported by the results, however when discussing the first theme the language surrounding taking the "middle-ground approach" did not make it clear that some doctors did take this approach (as mentioned in the results) so I would suggest revisiting this text to enhance clarity. Strengths and limitations are clearly described and the research's relevance is clear.

Reviewer #3: Conclusions are supported by the data and relevance to good clinical outcomes is discussed.

a. Pg. 17 

- line 360: “perspective” should be “perspectives”

- lines 372 and 373: It wasn’t only doctors who noted that tension in the consultation could be created by physicians telling patients not to use the complementary diet; Patient 13 said they “kept quiet” after the doctor told them not to drink the crab soup or papaya juice. It would have been interesting to hear what the patient thought about that specific physician - did they trust them? did they feel they were being listened to?

b. Good to see the authors address how patient stress can affect retention of information and that providing too much information during the visit can be overwhelming for the patient. The recommendation for physician training in communication is very appropriate.

c. Pg. 19

- line 411: The phrase “…made them understand better” is not clear nor an accurate summary of the two articles cited. Perhaps something like this clarifies the authors’ point: “The reason might be due to the stressful situation in the hospital that affected the patient’s reception and retention of information about the disease, whereas access to the information at home, away from the stressful hospital visit and feeling less fatigued allowed greater comprehension of the material.”

- lines 420-421: Suggest deleting the sentence “While doctors’ main considerations…” as it repeats the information from the previous sentence.

d. Pgs. 19 and 21: Suggest not using “unique” as the context is more individual rather than being the only one of a kind. For pg. 19 (line 430), perhaps “individual circumstances” and for pg. 21 (line 458), use of the word “challenges” without a descriptor would be appropriate given the context of home monitoring and self-care practices is specified.

**Editorial and Data Presentation Modifications?**

Reviewer #1: (No Response)

Reviewer #2: Propose a grammar check throughout, for example, tenses need looking at as they sometimes switch between present and past tense, singular language is also often used when the plural is needed.

Reviewer #3: Minor grammatical issues: 

1. The manuscript is well-written. However, a good read-through is needed to correct instances where a word should be plural but it is singular, and a very few that are plural but should be singular. Some examples are: 

- Abstract, lines 40 and 41: “doctor” should be “doctors” while “outpatients” should be “outpatient”

- Results, line 211: “benefit” should be “benefits”

- Results, line 258: “patient” should be “patients”

etc. 

2. Abstract, line 32: Word “patients” missing after “… targeting dengue…”

3. Introduction

a. Pg. 3, line 59: delete the extra “0” in the 32 per 1000,000 population

b. pg. 4, line 75: What does “default follow up” mean?

**Summary and General Comments**

Reviewer #1: Interesting results that with more power and evidence of conceptual overlap, could be of interest to Medical Practitioners within Singapore.

Reviewer #2: Overall, I thought this research was of extreme importance and believe it is novel in exploring dengue management in patients and doctors and identifying discrepancies between these populations thoughts surround health-seeking behaviour. I thought the narrative read really well - however, to make it even more well-rounded I would propose including in the introduction some information surrounding general dengue symptoms vs symptoms which are considered "more severe" to aid interpretation of the results from those who may be less familiar with dengue.

Reviewer #3: This is a nicely written manuscript on a topic that has not been deeply researched. The comparison of patient and physician perspectives on self-care practices and when to seek medical care for dengue fever is novel and helpful for planning health education interventions as strategies and materials need to be developed for both physicians and patients.

While the authors mention in the Discussion the need for communication training for physicians, this recommendation could be more strongly presented as the identified gaps between patients and physicians show an urgent need for better communication. A patient-centered care approach could be a useful framework for this recommendation. 

Additionally, some mention of who is responsible for developing interventions to address the incomplete knowledge about dengue, dengue warning signs in particular, and follow-up care would support the findings. For example, questions such as what combination of verbal and visual information should be shared with patients? How should this verbal and visual information be provided – during the visit, through print materials such as pamphlets or online videos, or a combination of methods, among others, should be addressed in a health education intervention.

PLOS authors have the option to publish the peer review history of their article (what does this mean?). If published, this will include your full peer review and any attached files.

Reviewer #1: No

Reviewer #2: No

Reviewer #3: No

Figure Files:

Data Requirements:

Reproducibility:

References

---

## [Editor Report · Decision Letter 1]

28 Feb 2023

Dear Dr. Ng,

Thank you very much for submitting your manuscript "Self-care practices and health-seeking behaviours in patients with dengue fever: a qualitative study from patients’ and doctors’ perspectives" for consideration at PLOS Neglected Tropical Diseases. As with all papers reviewed by the journal, your manuscript was reviewed by members of the editorial board and by several independent reviewers. The reviewers appreciated the attention to an important topic. Based on the reviews, we are likely to accept this manuscript for publication, providing that you modify the manuscript according to the review recommendations. 

Congratulations on addressing the the reviewer comments. Your article in quite interesting and demonstrate some important issues associated with patient/physician interactions. I work in Peru and see similar issues so I do think your finds have implications to areas beyond Malaysia. Your manuscript is much improved. 

That being said, I think your manuscript requires editing. It is understandable, but I think the style might be a bit off-putting to some readers (for example one of your original reviewers). I would be willing to do that editing, send the manuscript back to you with track changes and you could accept or reject what you want. I just want this manuscript to have the impact it deserves. I'm not sure what options you would have for editing, but I think I could do it in a day and want to support this manuscript. There are a few suggestions I would also like to make on some of the terminology you use.

Would it be appropriate say "use of traditional remedies" instead of "complementary diet". Related to this for readers in other parts of the world, could you provide some descriptions of what exactly "Papaya leaf extract" is and in what form it is ingested. Is it another drink from boiling papaya leaves, or is it a pill or supplement. Crab soup seems like crab soup. I work in Latin America and there are a host of local remedies that all amount to being liquids.

Honestly, I can't understand why a physician would discourage ingesting liquids (not sure about the papaya extract). I want to scold the physicians but if the papaya extract is a pill then it would clearly be optional. There are two issues here: This patients follow Drs advice about increasing fluid intake or not and if they decided to do it with traditionally recognized remedies. It is important to distinguish which involved ingesting liquids and which did not. How were the doctors making this distinction.

To give you a sample of my editing I'm attaching a track changes version edited through the introduction. See if you like my approach. If you do, you can keep the changes and try to make similar changes thorough out and I will edit the next version of your manuscript.

For the reviewer response, just give a few sentences address the "complementary diet" comments above and if you agree with my suggested edits and if you would like me to edit your next draft.

We fully intend to publish this manuscript, but again it needs editing. The sentence structure is a bit off, some errors with tense and plural agreement.

Sincerely,

Amy C. Morrison, PhD

Academic Editor

Elvina Viennet

Section Editor

Congratulations on addressing the the reviewer comments. Your article in quite interesting and demonstrate some important issues associated with patient/physician interactions. I work in Peru and see similar issues so I do think your finds have implications to areas beyond Malaysia. Your manuscript is much improved. 

That being said, I think your manuscript requires editing. It is understandable, but I think the style might be a bit off-putting to some readers (for example one of your original reviewers). I would be willing to do that editing, send the manuscript back to you with track changes and you could accept or reject what you want. I just want this manuscript to have the impact it deserves. I'm not sure what options you would have for editing, but I think I could do it in a day and want to support this manuscript. There are a few suggestions I would also like to make on some of the terminology you use.

Would it be appropriate say "use of traditional remedies" instead of "complementary diet". Related to this for readers in other parts of the world, could you provide some descriptions of what exactly "Papaya leaf extract" is and in what form it is ingested. Is it another drink from boiling papaya leaves, or is it a pill or supplement. Crab soup seems like crab soup. I work in Latin America and there are a host of local remedies that all amount to being liquids.

Honestly, I can't understand why a physician would discourage ingesting liquids (not sure about the papaya extract). I want to scold the physicians but if the papaya extract is a pill then it would clearly be optional. There are two issues here: This patients follow Drs advice about increasing fluid intake or not and if they decided to do it with traditionally recognized remedies. It is important to distinguish which involved ingesting liquids and which did not. How were the doctors making this distinction.

To give you a sample of my editing I'm attaching a track changes version edited through the introduction. See if you like my approach. If you do, you can keep the changes and try to make similar changes thorough out and I will edit the next version of your manuscript.

For the reviewer response, just give a few sentences address the "complementary diet" comments above and if you agree with my suggested edits and if you would like me to edit your next draft.

We fully intend to publish this manuscript, but again it needs editing. The sentence structure is a bit off, some errors with tense and plural agreement.

Figure Files:

Data Requirements:

Reproducibility:

References

---

## [Editor Report · Decision Letter 2]

26 Mar 2023

Dear Dr. Ng,

Thank you very much for submitting your manuscript "Self-care practices and health-seeking behaviours in patients with dengue fever: a qualitative study from patients’ and doctors’ perspectives" for consideration at PLOS Neglected Tropical Diseases. As with all papers reviewed by the journal, your manuscript was reviewed by members of the editorial board and by several independent reviewers. The reviewers appreciated the attention to an important topic. Based on the reviews, we are likely to accept this manuscript for publication, providing that you modify the manuscript according to the review recommendations. 

Hi All:

Attached is an edited manuscript. I'm glad I waited as you made many good changes in response to my suggestions. I did some heavy editing, but want to emphasize how much I like this paper. The more time I spend with it the more important I think it is.

Please read my edits and ensure I've not changed any meaning, especially in the discussion I may have gone a little to far.

Also take a last look at your discussion, to reduce it has much as possible. Although I think many of your suggestions are spot on they may detract from the key messages which are:

1. The health care system needs to rethink their messaging around traditional remedies.

2. Warning signs are not being effectively communicated to patients in this setting and this is a big problem.

3. We need new and creative ways to manage dengue cases, they do not exist yet, but research is needed sooner rather than later.

I highlighted some text in green - it is up to you but I think it can go.

If you are okay with all the suggested changes and you make your final decisions, this will be ready to go to production.

Thanks for being willing to work with us and for your patience, I was very busy the last two weeks and could not review until recently.

Best of luck and nice work.

All the best,

Amy

Sincerely,

Amy C. Morrison, PhD

Academic Editor

Elvina Viennet

Section Editor

Hi All:

Attached is an edited manuscript. I'm glad I waited as you made many good changes in response to my suggestions. I did some heavy editing, but want to emphasize how much I like this paper. The more time I spend with it the more important I think it is.

Please read my edits and ensure I've not changed any meaning, especially in the discussion I may have gone a little to far.

Also take a last look at your discussion, to reduce it has much as possible. Although I think many of your suggestions are spot on they may detract from the key messages which are:

1. The health care system needs to rethink their messaging around traditional remedies.

2. Warning signs are not being effectively communicated to patients in this setting and this is a big problem.

3. We need new and creative ways to manage dengue cases, they do not exist yet, but research is needed sooner rather than later.

I highlighted some text in green - it is up to you but I think it can go.

If you are okay with all the suggested changes and you make your final decisions, this will be ready to go to production.

Thanks for being willing to work with us and for your patience, I was very busy the last two weeks and could not review until recently.

Best of luck and nice work.

All the best,

Amy

Figure Files:

Data Requirements:

Reproducibility:

References

---

## [Editor Report · Decision Letter 3]

11 Apr 2023

Dear Dr. Ng,

We are pleased to inform you that your manuscript 'Self-care practices and health-seeking behaviours in patients with dengue fever: a qualitative study from patients’ and physicians’ perspectives' has been provisionally accepted for publication in PLOS Neglected Tropical Diseases.

Best regards,

Amy C. Morrison, PhD

Academic Editor

Elvina Viennet

Section Editor

Dear All:

Thanks so much for being willing to work with me. I am happy with and glad you did some rephrasing and think you have a very nice manuscript. I happen to be a bit passionate about the topic and hope this manuscript inspires additional studies on this topic. I have often told Health Policy Makers in Latin America where I work, that case management and appropriate communication of "Warning Signs" should be priority number one for dengue control program and that is coming from a vector biologist/control specialist. But like most things in Public Health it is not that simple. Congratulations!

Amy

---

## [Editor Report · Acceptance letter]

24 Apr 2023

Dear Dr. Ng,

We are delighted to inform you that your manuscript, "Self-care practices and health-seeking behaviours in patients with dengue fever: a qualitative study from patients’ and physicians’ perspectives," has been formally accepted for publication in PLOS Neglected Tropical Diseases.

Best regards,

Shaden Kamhawi

co-Editor-in-Chief

Paul Brindley

co-Editor-in-Chief
